# Research

ecology, evolution

local adaptation, source–sink dynamics, spatial sorting, heterogeneous habitats, spider mites, *Tetranychus urticae*

**Author for correspondence:**
Karen Bisschop
e-mail: kbisschop.evo@gmail.com

†Joint last authorship.

### PUBLISHING

# Transient local adaptation and source–sink dynamics in experimental populations experiencing spatially heterogeneous environments

Karen Bisschop[1,2], Frederik Mortier[2], Rampal S. Etienne[1,†] and Dries Bonte[2,†]

[1]Groningen Institute for Evolutionary Life Sciences, University of Groningen, PO Box 11103, 9700 CC Groningen, The Netherlands
[2]TEREC (Terrestrial Ecology Unit), Department of Biology, Ghent University, Karel Lodewijk Ledeganckstraat 35, 9000 Ghent, Belgium

KB, 0000-0001-7083-2636; FM, 0000-0002-1480-2675; RSE, 0000-0003-2142-7612; DB, 0000-0002-3320-7505

Local adaptation is determined by the strength of selection and the level of gene flow within heterogeneous landscapes. The presence of benign habitat can act as an evolutionary stepping stone for local adaptation to challenging environments by providing the necessary genetic variation. At the same time, migration load from benign habitats will hinder adaptation. In a community context, interspecific competition is expected to select against maladapted migrants, hence reducing migration load and facilitating adaptation. As the interplay between competition and spatial heterogeneity on the joint ecological and evolutionary dynamics of populations is poorly understood, we performed an evolutionary experiment using the herbivore spider mite *Tetranychus urticae* as a model. We studied the species's demography and local adaptation in a challenging environment that consisted of an initial sink (pepper plants) and/or a more benign environment (cucumber plants). Half of the experimental populations were exposed to a competitor, the congeneric *T. ludeni*. We show that while spider mites only adapted to the challenging pepper environment when it was spatially interspersed with benign cucumber habitat, this adaptation was only temporary and disappeared when the populations in the benign cucumber environment were expanding and spilling-over to the challenging pepper environment. Although the focal species outcompeted the competitor after about two months, a negative effect of competition on the focal species's performance persisted in the benign environment. Adaptation to challenging habitat in heterogeneous landscapes thus highly depends on demography and source–sink dynamics, but also on competitive interactions with other species, even if they are only present for a short time span.

## 1. Background

Local adaptation is a major driver of range expansion and invasion [1]. Species that colonize new areas outside their native range are likely to end up in novel, often heterogeneous environments. The persistence and establishment of populations in these new environments does not only depend on the number of founders but also on the local environmental filter, which may be overcome by adaptation [2,3]. Moreover, persistence will be facilitated by fitness stabilizing mechanisms following resource partitioning where individuals or species specialize in either a genetic or plastic way on different resources in a patch [4,5].

The distribution of species is not restricted to areas where expected fitness is positive. While dispersing, species also come into contact with marginal and

even completely unsuitable habitat, but may reach substantial local population sizes there via spill-over effects. Species may therefore be ecologically rescued in marginal habitat by source–sink dynamics [6–8]. Dispersal towards these harsher habitats may be beneficial to escape strong competition with superior individuals or species. This will equalize fitness across habitats and promote regional coexistence [4,9]. These habitats differ not only in their local conditions but also in their connectedness [10], rendering insights from source–sink theories useful to understand demography. The extent of fragmentation as reflected by the number, size and connectedness of habitats will strongly affect dispersal and drift, and therefore local adaptation [11,12]. Connectedness may facilitate this local adaptation if standing genetic variation is maintained in connected but less challenging habitats (i.e. under moderate dispersal, sink populations may undergo evolutionary rescue). However, too much gene flow is known to hamper adaptation (migration load [13,14]). In these situations, persistence is more likely due to ecological rescue alone [6,15]. Intermediate amounts of dispersal are, therefore, thought to be most beneficial for evolutionary rescue [16], depending on the dispersal-selection balance [17].

Ecological rescue by dispersal from source populations is, in the long run, unstable unless further adaptation occurs [18–20]. This evolutionary rescue is more successful when the stressors imposed by the habitat are organized along gradients [21]. Such gradients result in shallower selection clines, and hence, maintain the genetic variation needed to fuel further adaptations along the gradient. The presence of moderately challenging habitat is especially beneficial for persistence if general, rather than specialized stress responses can evolve [22], such as when there is selection for phenotypes that are able to deal with a broad array of the stressor and/or perform well in several habitats (generalism) [23]. An example of a general stress response is fluctuating temperatures leading to thermal generalism [23].

Species usually colonize environments that are inhabited by other species. The presence of heterospecific competitors is known to complicate the processes of local adaptation, species persistence and coexistence [24]. Species interactions will affect the strength of the local selection pressures, thereby accelerating or decelerating adaptation rates [14,25–27]. Higher adaptation rates are expected when selection pressures imposed by competition overcome the negative effects of lower population sizes as long as traits for adaptation to new habitats and for higher competitive fitness are aligned [26]. Even species that are driven to extinction—these are called ghost species—can have a long-term evolutionary impact because they may create habitat modifications or evolutionary changes during their episodes of existence in the community [28–30].

Environmental heterogeneity and competition are thus anticipated to affect the process of local adaptation to novel habitats, but to our knowledge, they have not been simultaneously studied experimentally. Here, we present the results of an evolutionary experiment in which the two-spotted spider mite (*Tetranychus urticae*) is transferred towards homogeneous or mixed sets of host species with or without the congeneric competitor, *T. ludeni*. We test the prediction that adaptation to the challenging pepper plants is more likely in the heterogeneous environment by evolutionary and ecological rescue. We furthermore expect interspecific competition to promote the exploitation of challenging resources and consequently to lead to faster adaptation to the challenging environment. We here confirm these predictions but highlight the non-trivial pathways leading to persistence in marginal habitats.

## 2. Methods

### (a) Study species
The species for this study are members of the family Tetranychidae (Acari, Arachnida): the two-spotted spider mite, *Tetranychus urticae* Koch (focal species), and the red-legged spider mite, *Tetranychus ludeni* Zacher (congeneric competitor). Large populations can be easily maintained due to their small body sizes (approx. 0.4 mm). Their short generation times, high performance, fast responses to selection and high levels of standing genetic variation [31–36], render spider mites highly suitable for experimental evolution.

### (b) Experimental set-up
The experimental set-up is visualized in electronic supplementary material, figure S1. We created 13 inbred lines of *T. urticae* by mother–son mating. These lines were already highly inbred from previous experiments [37]. The initial population for these inbred lines was the LS-VL line, which started from about 5000 spider mites collected from roses in October 2000. They were afterwards maintained on bean plants, *Phaseolus vulgaris* 'Prelude'. We tried to create 13 inbred lines for the competitor (*T. ludeni*) as well, but we failed because of low fertility and early mortality. Therefore, we created only six inbred lines of the competitor, which were also kept on *Phaseolus vulgaris* 'Prelude'. We chose these strongly inbred lines to create similar gene pools for the evolutionary experiment and thereby reduce variability in outcomes simply due to variability in starting genetic variation.

We were interested in the effect of spatial heterogeneity and the impact of an interspecific competitor on adaptation to novel environments. Environmental variation was created by using different host plants, which served as resources to the mites. To investigate the effect of the resources, we created independent experimental units (islands) that each contained several plants among which dispersal was possible. Islands consisted of four three-week-old cucumber plants, *Cucumis sativus* 'Tanja', four five-week-old pepper plants, *Capsicum annuum* 'California Wonder', or a mixture with two cucumber and two pepper plants. All plants within the islands were in direct contact with each other. Cucumber is the more palatable of the two and hence hypothesized to present an ecological source and a possible evolutionary stepping stone to adaptation to pepper. To prevent spider mites from crossing between islands, the bottom of the boxes with plants was covered with yellow sticky paper (Pherobank) and the walls were covered with Vaseline. This method was effective in previous work [14]. To examine the impact of the competitor, 24 islands received only *T. urticae* (eight replicates for each treatment: homogeneous cucumber, homogeneous pepper and the mixture), and another 24 also received the heterospecific *T. ludeni*. We initiated each island with 52 adult females. Islands without interspecific competition received four adult females from each of the 13 inbred lines of our focal species, and islands with interspecific competition received two adult females from the same inbred lines, supplemented with 26 adult females of the competitor, *T. ludeni*. Twelve of the *T. ludeni* females were from the six successful inbred lines, while the 14 others were taken from the stock population that was kept on bean. The initial population size of 52 adult females was chosen because natural populations

colonize plants at low population sizes and it was found that inbreeding does not influence genetic trait variation in *T. urticae* [38]. We created a control population on bean that was initiated with the same isogenic lines from *T. urticae*.

Every week, we assessed the deterioration of the plants in the islands. When necessary, the two oldest plants were replaced by two new plants. This way of refreshment guaranteed enough time for the mites to move towards the fresh plants and also for a generation of mites to develop. Although half of the plants per island were removed, this was much less than half of the population as most mites did not prefer to stay on the deteriorated plants. Therefore, this replacement protocol was preferred over fixed intervals regardless of the state of deterioration. The temperature in the climate-controlled room was 25–30°C and the light regime was 16 : 8 LD. For logistical reasons, we performed the experiment in two blocks with four replicates per treatment (hereafter 'blocks'). The experiment lasted for 10 months which is equivalent to about 25 generations, considered to be sufficiently long to detect local adaptation in this species [33–36,39].

## (c) Measurements

For the ecological dynamics, the density of mites per unit surface was tracked. Density of individuals per unit of resource is a better representation of the current competition than total population size. We assessed the density on the islands every two weeks by counting the number of adult females on a 1 cm$^2$ square next to the stalk of the highest fully grown leaf of the newest plants of the island. A specific location standardizes the measurements and enables a more reliable comparison in time. Both the abaxial and the adaxial side of the leaf were measured and the numbers summed.

We assessed evolutionary dynamics in performance by measuring fecundity. We chose fecundity as a proxy of adaptation because previous research demonstrated it to be the best predictor of adaptation compared to survival or development [14,35,40]. From here on we refer to 'fecundity' and 'performance' interchangeably. We sampled five *T. urticae* females from each plant species per island at 2, 4, 6, 8 and 10 months during the experiment. We placed them on bean leaf discs for two generations to standardize juvenile and maternal effects [41,42]. These bean leaf discs (17 × 27 mm$^2$) were surrounded by paper strip borders in Petri dishes with wet cotton wool. From this last generation, three quiescent deutonymph females were taken and placed separately each with an adult male from the same leaf on, respectively, a bean, cucumber and pepper leaf (same set-up as for common garden) in a climate cabinet at 30°C. The number of eggs and larvae after 6 days was assessed with daily pictures (hereafter referred to as 'performance') and females that drowned before the 6th day were excluded from the analysis.

## (d) Statistical analysis

We first computed the goodness-of-fit of various parametric distributions to our performance data, separately for each plant species on which the performance was assessed. Based on these results, we chose a Gaussian distribution for the data from bean and cucumber, while the best-fitting distribution for performance on pepper was a zero-inflated negative binomial distribution with log link function, because of overdispersion in these data. The variance was determined as μ*k* in which μ is the mean and *k* is the overdispersion parameter (linear-variance parameterization). We chose a single zero-inflation parameter because the overall percentage of zeroes per month in the dataset did not change in time. We tested this with a linear model (95% confidence interval = [60.83; 79.47]; $Z = 1.268$; $p = 0.205$). These distributions were used in generalized linear mixed models (GLMMs).

To control for changes in performance caused by different leaf qualities between assessed time points, we investigated differences in performance between the control population and the experimental populations through time for each plant species (bean, cucumber and pepper). We expected no differences in performance through time for the control population because they have always been maintained on bean plants. The dependent variable was performance (i.e. fecundity, number of eggs laid after 6 days), and the fixed explanatory variables were time, main treatment (control or experiment, i.e. both pepper and cucumber) and the time-by-treatment interaction. Because the replicates were split into two blocks for logistical reasons, we nested the islands in blocks that were treated as a random variable in the statistical model. Model selection was based on the lowest AICc and the pairwise comparisons of the least square means were adjusted for multiple comparisons based on Tukey's method. For an overview of the importance of the separate independent variables and their interactions, we also performed Wald χ$^2$ tests of the maximal model. Time was treated as a categorical variable (2, 4, 6, 8 and 10 months) rather than a continuous variable, because we cannot assume a linear response of adaptation and because leaf quality for the fecundity tests might change in time.

To determine the effect of interspecific competition and habitat heterogeneity on adaptation through time, we built a GLMM per plant species with performance (number of eggs laid after 6 days) as the dependent variable. The fixed explanatory variables in the full model were time, treated as a categorical variable (2, 4, 6, 8 and 10 months), interspecific competition, also treated as a categorical variable (the presence or absence of *T. ludeni*), and the combination of the homogeneity or heterogeneity of the island with the plant where the females were sampled from (homogeneous from cucumber, heterogeneous from pepper and heterogeneous from cucumber). The populations on the homogeneous pepper islands were not able to survive, so they were not considered. The random variables were the different islands nested within the two blocks. Model selection was based on the lowest AICc and Wald χ$^2$ tests were performed on the maximal model. The pairwise comparisons of the least square means were adjusted for multiple comparisons based on Tukey's method.

A final test was performed in which we were interested in potential differences between the plant species within the heterogeneous islands. The model included performance as the dependent variable and the fixed effects were the plant species (cucumber or pepper) and the time (2, 4, 6, 8 and 10 months), both treated as categorical variables. The different islands nested within the blocks were random variables. We performed Wald χ$^2$ tests of the independent variables of the maximal model. Least square means between the different plant species were computed in which the *p*-values were adjusted for multiple comparisons based on Tukey's method.

The estimates provided in the tables are the raw and untransformed estimates for the fixed effects of the final models, of which the one for performance on pepper has a negative binomial error distribution.

All analyses were performed in R (v. 3.5.1) with glmmTMB v. 0.2.2.0 [43], MuMIn v. 1.42.1 [44], emmeans v. 1.2.4 [45], fitdistrplus v. 1.0-11 [46] and tseries v. 0.10-45 [47].

## 3. Results

Our main findings are fourfold. (i) Mites adapted to the most benign habitat (cucumber) in both homogeneous and heterogeneous environments. (ii) Populations in the sink habitat (pepper) established in heterogeneous environments only. (iii) Temporary local adaptation in the sink habitat (pepper) occurred within heterogeneous environments. (iv) Competition

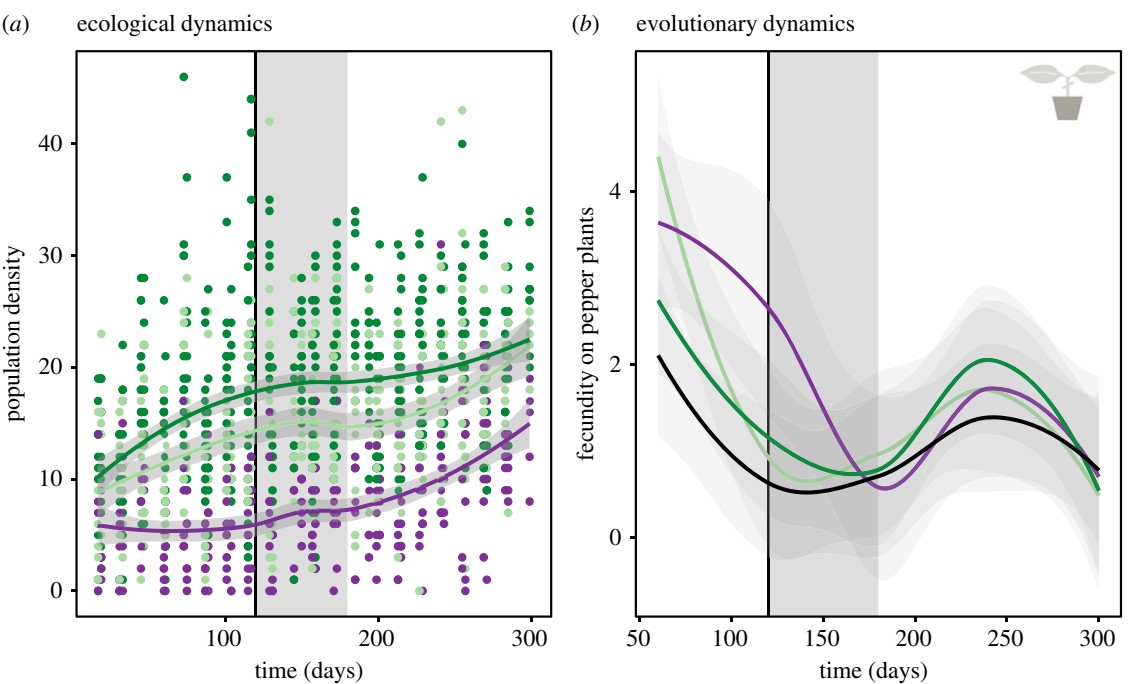

**Figure 1.** Evolutionary and ecological dynamics. (*a*) Population densities (i.e. adult females per cm²) on homogeneous cucumber plants (dark green), heterogeneous cucumber (light green) and pepper (purple) plants. (*b*) Changes in performance (i.e. number of eggs after 6 days) tested on pepper plants per time point. The different colours represent mites taken from the homogeneous cucumber plants (dark green), the heterogeneous cucumber plants (light green), heterogeneous pepper plants (purple) or bean plants (control; black). In both panels, the black vertical line indicates the four months mark, where a fecundity test was performed and a significantly higher performance of mites taken from pepper compared to cucumber was found. The grey zone shows the time between the assessments on four and six months. (Online version in colour.)

negatively affected performance on the ancestral habitat (bean), but not on the novel habitats (cucumber, pepper). Below we detail these findings.

## (a) Ecological dynamics

In the heterogeneous islands, the focal species's population density on cucumber increased immediately, while the density on pepper initially remained stable at about five adult females per cm². After two months—the density on cucumber had already reached about 15 adult females per cm²—a steep increase was observed on both plant species (figure 1*a*). Mites reached higher densities in the homogeneous islands more rapidly, but the densities levelled off to similar values as on the heterogeneous islands (figure 1).

The populations of our focal species, *Tetranychus urticae*, and the competitor, *T. ludeni*, on the homogeneous pepper islands went extinct after a few weeks. Therefore, we were unable to perform fecundity tests on populations from the homogeneous pepper islands. Pepper habitat thus represented a true sink habitat where neither species could persist. On the homogeneous cucumber islands and the heterogeneous islands, the interspecific competitor, *T. ludeni*, could not survive and went extinct after about two months.

## (b) Ancestral versus novel host plants

We detected a signal of local adaptation for the populations on cucumber; populations taken from the novel host plants (cucumber and/or pepper) evolved a 10 per cent higher fecundity on the cucumber leaves than the control population (kept on bean), independent of time ($Z = 2.544$ and $p = 0.011$; figure 2*b*; table 1; electronic supplementary material, tables S1 and S2). When assessed on bean leaves, populations from the

novel host plants (cucumber and/or pepper) showed an overall performance similar to the control population, indicating no loss of adaptation to the original host (bean). In fact, they even reached higher fecundity (an increase of 16 per cent) than the control at four months ($Z = 2.266$ and $p = 0.0234$; figure 2*a*; table 1; electronic supplementary material, tables S1 and S2). On pepper, the populations from the novel environments (cucumber and/or pepper) performed as badly as the control population, indicating no local adaptation to the pepper plants (figure 2*c*; table 1; electronic supplementary material, tables S1 and S2).

## (c) Interspecific competition and time

We found an effect of time and interspecific competition on the performance measured on bean for the experimental populations (figure 3*a*; table 2; electronic supplementary material, tables S1 and S2). All populations, including the control population, had a lower performance in the last month, which seems to be due to differences in the quality of leaves used in the fecundity tests. Mites under interspecific competition laid significantly fewer eggs on bean, a decrease of eight per cent, than those from the treatment with only intraspecific competition ($Z = 2.901$; $p = 0.00371$; table 2; electronic supplementary material, tables S1 and S2). These relative differences in performance are probably due to drift effects caused by a lower number of founders in the islands with interspecific competition than in those with only intraspecific competition. The presence of interspecific competition was the only explanatory variable in the best model on cucumber, but the difference between both treatments was not significant ($Z = 1.51$; $p = 0.132$; figure 3*b*; table 2; electronic supplementary material, table S1) and just not

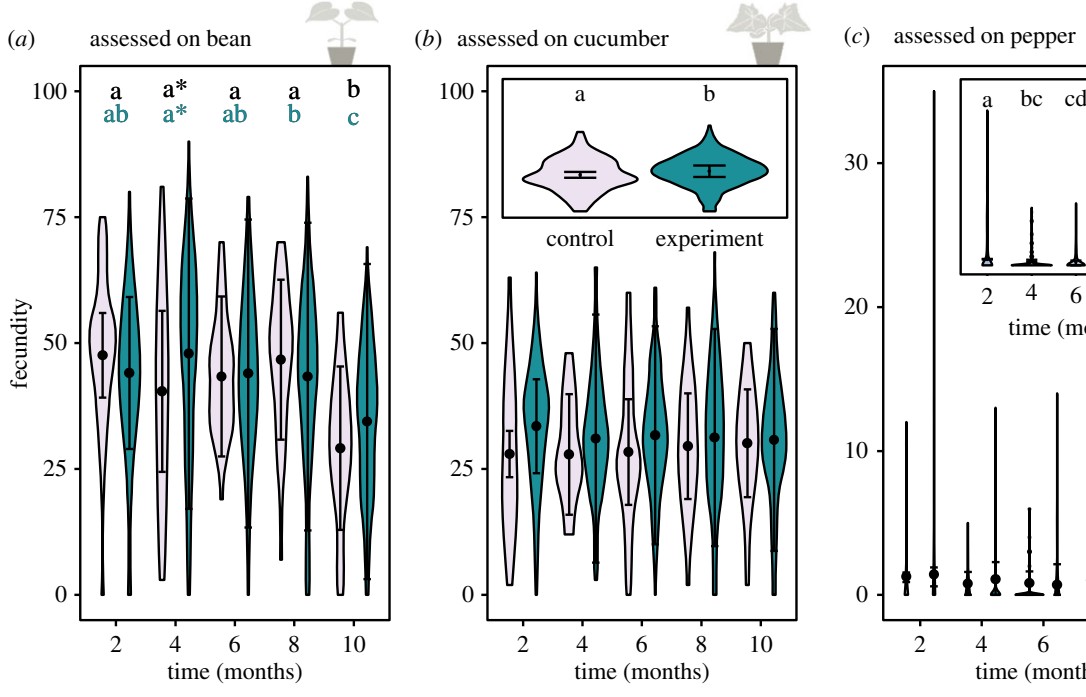

**Figure 2.** Evolutionary dynamics for control and experimental populations: changes in performance (i.e. number of eggs after 6 days). (*a*) Assessed on bean. Time, origin (control or experiment) and their interaction significantly affect performance. The black letters indicate significant differences in time for the control and in blue for the experiment; the asterisk shows the difference between both treatments. (*b*) Assessed on cucumber. Only treatment affects performance significantly, which is visualized in the inserted plot). (*c*) Assessed on pepper. Only time has a significant effect on performance as indicated in the inserted plot. The violin plots show the observed data, while the points and lines show the mean model estimates and their 95% confidence interval, respectively. (Online version in colour.)

**Table 1.** Summary of the final GLMM explaining total performance for comparison of the experimental and ancestral population in time. (*a*) Fecundity assessed on bean: the final model included time, the origin of the mite (experiment or control) and their interaction. (*b*) Fecundity assessed on cucumber: mites from the experiment performed significantly better. (*c*) Fecundity assessed on pepper: no difference between control and experimental populations was found, only time had a significant negative effect.

| | | estimate | s.e. | Z-value | Pr (>\|z\|) |
|---|---|---|---|---|---|
| (*a*) bean | (intercept) | 47.5620 | 4.2964 | 11.070 | $< 2 \times 10^{-16}$*** |
| | four months | −7.1388 | 3.8591 | −1.850 | 0.0643 |
| | six months | −4.2116 | 3.8049 | −1.107 | 0.2683 |
| | eight months | −0.8783 | 3.8049 | −0.231 | 0.8174 |
| | 10 months | −18.4544 | 3.9770 | −4.640 | $3.48 \times 10^{-16}$*** |
| | Experiment | −3.5087 | 3.3864 | −1.036 | 0.3001 |
| | four months : exp. | 10.9829 | 4.1670 | 2.636 | 0.0084** |
| | six months : exp. | 4.1284 | 4.1055 | 1.006 | 0.3146 |
| | eight months : exp. | 0.1652 | 4.0926 | 0.040 | 0.9678 |
| | 10 months : exp. | 8.8113 | 4.2952 | 2.051 | 0.0402* |
| (*b*) cucumber | (intercept) | 28.869 | 1.211 | 23.846 | $<2 \times 10^{-16}$*** |
| | experiment | 2.889 | 1.136 | 2.544 | 0.011* |
| (*c*) pepper | (intercept) | 1.9886 | 0.1101 | 18.063 | $< 2 \times 10^{-16}$*** |
| | four months | −0.8767 | 0.1926 | −4.552 | $5.32 \times 10^{-6}$*** |
| | six months | −1.4587 | 0.2074 | −7.035 | $1.99 \times 10^{-12}$*** |
| | eight months | −0.6534 | 0.1529 | −4.274 | $1.92 \times 10^{-5}$*** |
| | 10 months | −1.5919 | 0.2145 | −7.421 | $1.17 \times 10^{-13}$*** |

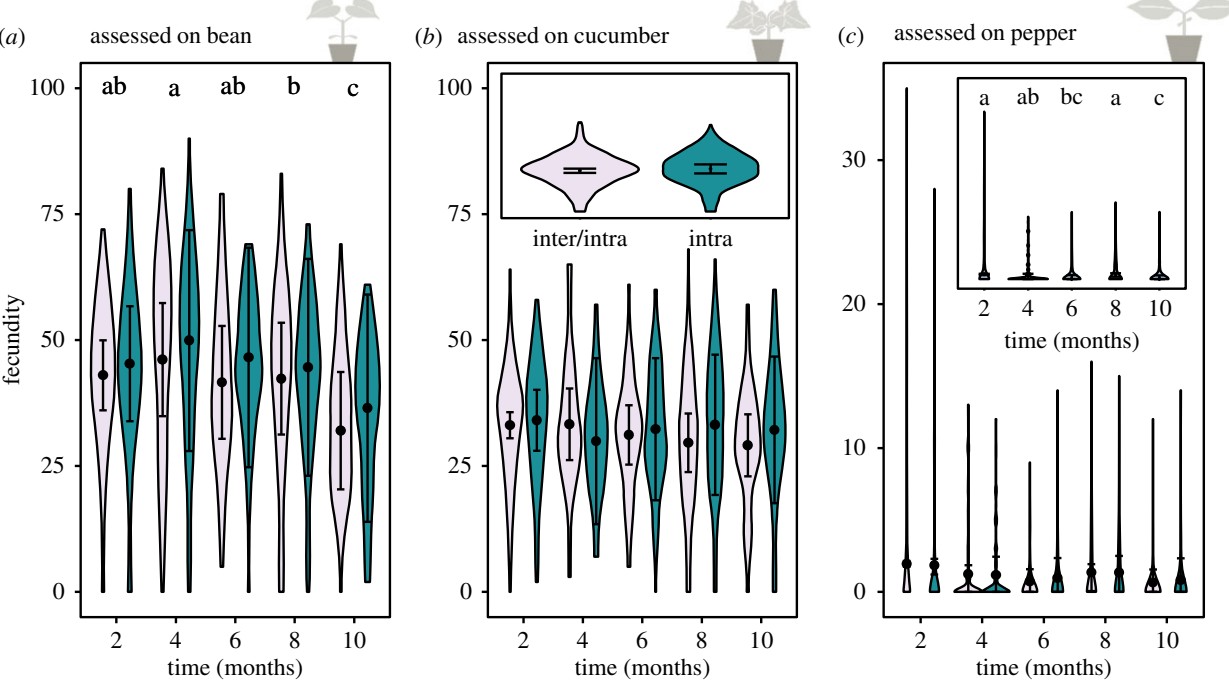

**Figure 3.** Evolutionary dynamics for experimental populations. Changes in performance (i.e. number of eggs after 6 days). (*a*) Assessed on bean. The time and the presence or absence of interspecific competition significantly affect performance. The black letters indicate significant differences in time. Mites under interspecific competition had a significantly lower performance. (*b*) Assessed on cucumber. Only competition was included in the final model, which is visualized in the inserted plot. (*c*) Assessed on pepper. Only time has a significant effect on performance as indicated in the inserted plot. The violin plots show the observed data, while the points and lines show the mean model estimates and their 95% confidence interval, respectively. (Online version in colour.)

**Table 2.** Summary of the final GLMM explaining the total performance of the experimental populations (including heterogeneous and homogeneous populations). (*a*) Fecundity assessed on bean: time significantly influenced the performance and the absence of *T. ludeni* had a significant positive effect on performance. (*b*) Fecundity assessed on cucumber revealed no significant differences. (*c*) Fecundity assessed on pepper showed a significant effect of time.

| | | estimate | s.e. | *Z*-value | *Pr* (>\|z\|) |
|---|---|---|---|---|---|
| (*a*) bean | (intercept) | 42.4354 | 3.4129 | 12.434 | $<2 \times 10^{-16}$*** |
| | no competition | 3.4794 | 1.1992 | 2.901 | 0.00371** |
| | four months | 3.8220 | 1.5859 | 2.410 | 0.01595* |
| | six months | −0.1068 | 1.5565 | −0.069 | 0.94531 |
| | eight months | −0.6940 | 1.5215 | −0.456 | 0.64831 |
| | 10 months | −9.8571 | 1.6390 | −6.014 | $1.81 \times 10^{-9}$*** |
| (*b*) cucumber | (intercept) | 31.0613 | 0.8459 | 36.72 | $<2 \times 10^{-16}$*** |
| | no competition | 1.4345 | 0.9531 | 1.51 | 0.132 |
| (*c*) pepper | (intercept) | 0.9428 | 0.1491 | 6.323 | $2.56 \times 10^{-10}$*** |
| | four months | −0.4258 | 0.1771 | −2.404 | 0.0162* |
| | six months | −0.9496 | 0.1943 | −4.887 | $1.02 \times 10^{-6}$*** |
| | eight months | −0.1068 | 0.1545 | −0.691 | 0.4896 |
| | 10 months | −1.1249 | 0.2202 | −5.108 | $3.26 \times 10^{-7}$*** |

significant in the $\chi^2$ test ($p = 0.089$; electronic supplementary material, table S2).

Fecundity on pepper decreased in time and seemed to coincide with population build-up (figure 3*c*; table 2; electronic supplementary material, tables S1 and S2), which is similar to the control population. The observed performance on pepper was lowest after 10 months. No effect of interspecific competition on pepper was found.

### (d) Comparison between mites taken from cucumber and pepper in heterogeneous islands

Overall, we did not find significant differences between mites sampled from the cucumber and pepper plants within the same heterogeneous islands for performance on the initial host plant, bean, and the novel host plant, cucumber. However, we did detect a transient signal of local adaptation. At four months, mites sampled from pepper plants had a significantly higher fecundity on pepper, an increase of 44 per cent, than the mites sampled from cucumber from the same island ($Z = -2.546$; $p = 0.0109$; figure 1b). This signal vanished after six months. By contrast, mites sampled from cucumber did not perform better on cucumber than mites sampled from pepper at any point in time.

## 4. Discussion

We have shown that the establishment of populations in novel environments depends on the tight interplay between transient ecological and evolutionary dynamics in response to the nature and diversity of the local conditions (here host plant identity). Establishment in the most challenging environment (pepper) was impossible when it was the only available one. However, when this marginal habitat was spatially interspersed with a more benign one (cucumber), establishment succeeded because rapidly expanding populations on the neighbouring benign habitat generated sufficient immigration. These source–sink dynamics thus initially created ecological rescue in the challenging environment. Evolutionary rescue (i.e. adaptation to the challenging environment) was, however, only transient. We speculate that this is because a further expansion of the source populations broke down adaptation through an influx of unadapted genes (i.e. genetic load).

Although transient, the signal of adaptation in our heterogeneous habitat was convincing: mites that were sampled from pepper plants reached the highest performance on that host plant. The difference in performance on the two host plants suggests that dispersal was not random. A possible explanation is isolation by environment through biased dispersal [4,48], as it is known that dispersal with habitat choice can favour rapid evolution [49]. Also, non-random dispersal might result from a competition–colonization trade-off, as suggested by earlier dispersal experiments with this model species [50]. The general theory states that competitively inferior individuals may be better at colonization [51], which could be consistent with our observations, as individuals on the more challenging pepper plants could be escaping competition on the less challenging and, therefore, more densely populated cucumber plants. Also, when negative frequency dependence is operating, invading lower population sizes could be beneficial if invaders are ecologically distinct. This non-random dispersal probably vanished after the first months and it was replaced by random spill-over from cucumber to pepper.

Overall, and independently of the exact mechanism, establishment on the less challenging cucumber habitat was essential as homogeneous pepper populations became extinct. Therefore, we argue that the neighbouring populations on cucumber served as evolutionary stepping stones for evolutionary and ecological rescue [18]. The theory behind evolutionary stepping stones is illustrated by Bell & Gonzalez [21]: gradual adaptation maintains enough genetic variation for further adaptation. While probably a common mechanism in range expansions and invasions, empirical evidence is rare: Fitzpatrick et al. [20] performed an experiment with translocations of Trinidadian guppies where they investigated the impact on downstream native populations. They found clear evidence that even low levels of gene flow from different ecotypes can assist small populations through ecological and evolutionary rescue.

The level of dispersal changed during the course of the experiment. This was demonstrated by the steep growth in population density on pepper plants tracking the growth in population density in the source. Source–sink dynamics (sensu [9]) arose, in which the cucumber plants served as sources with a strongly positive ratio of birth and immigration relative to death and emigration, while the pepper plants functioned as sinks. This indicates that the initial adaptation to the challenging host plant through resource partitioning was counteracted by increasing levels of dispersal from the source. The enlarged dispersal occurred at a density of ten to fifteen adult females per square centimetre which is the same threshold found by Bitume et al. [37]. This is in line with both theoretical predictions and some empirical work: local adaptation strongly depends on dispersal rates and limited gene flow is favoured [7,14,40].

Interestingly, while the competing species was driven to extinction after about two months, we still found a negative effect on performance on the initial host plant after 10 months. It is surprising that we found no significant differences in the effects of competition on fecundity assessed on cucumber and pepper plants, as we expected maladaptation to new hosts to increase the effects of interspecific competition. Even though the competitor went extinct, ghost interactions are known to affect performance by a reduction in effective population size, and hence an increase in drift effects [52]. However, the differences in initial population sizes between both treatments could also have caused strong drift effects, thus overriding any signal from competition.

## 5. Conclusion

Our study demonstrates transient adaptation in colonization processes with episodes of higher potential for adaptation to extreme environments via a combination of ecological and evolutionary rescue. Persistent adaptation was however never achieved, presumably because it was eventually completely overruled by spill-over. While in a homogeneous challenging habitat none of the populations was able to establish, in heterogeneous habitats establishment did occur: less challenging habitat can host source populations that play an important role as evolutionary stepping stones for further adaptation to more challenging habitats, by generating moderate dispersal. Furthermore, we emphasize that too much dispersal is disadvantageous because it counteracts ongoing local adaptation, and that interspecific interactions impact these dynamics even beyond their extinction. To maintain biodiversity in our changing world, it is of utmost importance to understand which factors affect local adaptation. As populations are not able to establish in homogeneous landscapes consisting of marginal habitat, persistence can be ensured in intermediately connected landscapes where

some patches of good quality may serve as enhancers of local adaptation to more marginal habitat.

Data accessibility. The data and script can be found at: https://hdl. handle.net/10411/PLHOGQ.

Authors' contributions. The idea and the design of the study were developed by K.B., D.B. and R.S.E. The data were collected by K.B. and F.M. and the statistical analyses were done by K.B. K.B. wrote the draft of the manuscript and all authors contributed to revisions and discussions.

Competing interests. We declare we have no competing interests.

Funding. D.B. and R.S.E. are supported by the FWO network EVENET (W0.003.16N) and FWO research grant no. G018017N. R.S.E. thanks the Netherlands Organisation for Scientific Research (NWO) for financial support through a VICI grant (VICI grant no. 865.13.00). K.B. thanks the Special Research Fund (BOF) of Ghent University and the Ubbo Emmius sandwich program of the University of Groningen

Acknowledgements. We thank Viki Vandomme, Angelica Alcantara, Pieter Vantieghem, Katrien Van Petegem, Stefano Masier, Matti Pisman, Mike Creutz, Hilde De Nil and Johan Bisschop for making the experiments possible. We also thank Raphaël Scherrer for the comments on a previous version of the manuscript and Sarah Magalhães for providing the strains of *T. ludeni*. We thank the Terrestrial Ecology department of Ghent University and the Centre for Ecology, Evolution and Environmental Changes of the University of Lisbon for the spider mite populations.

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
