## [Reviewer comments · Proceedings of the Royal Society B: Biological Sciences]

Review History

RSPB-2019-0135.R0 (Original submission)

Review form: Reviewer 1

Recommendation

Major revision is needed (please make suggestions in comments)

Scientific importance: Is the manuscript an original and important contribution to its field?

Good

General interest: Is the paper of sufficient general interest?

Good

Quality of the paper: Is the overall quality of the paper suitable?

Good

Is the length of the paper justified?

Yes

Should the paper be seen by a specialist statistical reviewer?

No

Do you have any concerns about statistical analyses in this paper? If so, please specify them explicitly in your report.

Yes

It is a condition of publication that authors make their supporting data, code and materials available - either as supplementary material or hosted in an external repository. Please rate, if applicable, the supporting data on the following criteria.

Is it accessible?

N/A

Is it clear?

N/A

Is it adequate?

N/A

Do you have any ethical concerns with this paper?

No

Comments to the Author

In this ms, the authors use a lab-based experimental-evolution system to investigate how habitat heterogeneity (in quality and diversity) and competitive interactions can potentially interact to shape local adaptation in a meta-community context. They interpret their results to suggest that in heterogenous landscapes, adaptation towards stressful habitats may be critically shaped by both ecological dynamics via the contribution of benign habitats acting as source habitats amidst patches of sink habitats, as well as transient competitive and adaptive effects.

Overall comments:

The questions posed here are excellent and very much at the cutting edge of unresolved questions at the interface of ecology and evolutionary biology. Moreover, I really enjoyed reading this paper and it got me thinking about a number of issues in evolutionary ecology in a meta-population context.

However, I feel that the current version of the ms does not allow for a full evaluation of the results and that a number of additional issues may need to be considered further. Detailed comments are below. Overall, I felt that the presentation of the results needs to be re-worked and the decisions for the statistical analyses need to be further justified. At times, some of the results seem to be cherry-picked with the Discussion focusing on marginal effects that do not seem justified enough to be the focus. Additionally, the current presentation of the results relies too much on statistical significance rather than presentations of effect sizes. The observed effect sizes seem rather marginal and while statistically significant, do not appear biologically very meaningful – although to be fair I think the authors can address this given their knowledge of the study system and certainly even marginal increases in fitness are crucial for adaptive evolution. I do see merit in this study and found it fascinating, but I do feel that some work is needed to be able to better evaluate the work done here.

I hope that my comments are helpful to the authors in revising the presentation of their very interesting study.

Minor comments:

Line 61: Why resource partitioning? Seems that any environmental factor that could give a species a negative-frequency dependent advantage (e.g., resources, predators, mutualists) could be equally important.

Lines 64: Unclear what is meant by a sink becoming “preferred.” I can see how they could still be used, but how would a preference for them evolve through selection? Selection does not move populations down fitness surfaces.

81: I’d specify locally stabilizing mechanism, because regionally it could be stabilizing.

93: This is not universally true and would only occur if traits conferring competitive fitness advantages are aligned in the direction of adaptive evolution for traits driving local adaptation to some new habitats.

195: What are the bracketed values?

207 and 221: I do not understand the rationale for a model selection procedure here. The goal does not seem to be to predict performance per se or to determine what factors contribute most to performance. Rather, the goal is to test a hypothesis that a set of factors and their interactions can have an effect on performance. As such, I would prefer to see the full model for any comparisons. There seems to be no need for model selection procedures in this study. I might be misunderstanding this, but the full two (intra and interspecific competition) by four (control, pepper, cuk, pepp/cuk) factorial model design needs to be implemented in the statistical model and those results presented. Planned contrasts between the core comparisons could then be done.

211: I do not see how intraspecific competition per se was assessed here. To do so would require manipulating density and showing that as density increases fecundity declines. Perhaps this is known from previous experiments or other work in this system, but it is critical to demonstrate competition, rather than simply assuming it. Perhaps the authors simply mean effect of interspecific competition.

218: Unless there is a gross issue with non-linearity in plant deterioration rates, treating time as a category and the resulting many multiple comparisons among time categories, rather than a slope seems odd to me. It seems that knowledge of the study system should dictate this “possible” non-linear effect among the plant species.

241-: I realize that there was extinction on the pepper plant, but unless this occurred immediately, it would be useful to see the same figure as figure one, but for the homogeneous treatments as well; e.g., Figure 1B should include the homogenous treatments. This would allow for a nice visualization. Figure S1 is close and should be moved to the main text, but include the homo pepper as well. This would also provide an illustration for the points made on Lines 249-255.

In figure 1, why does time begin at 50 days? Why not at the start of the introductions (e.g., day 0)?

Note: For all figures – please avoid red and green for choice of colors. Quite a few folks are color blind and these colors cannot be distinguished.

Shouldn't figure 2 y axes be labelled fecundity as well?

254: It's confusing to call one taxa an "interspecific competitor" because they are both interspecific competitors. Please state which went extinct.

258: Please give some measures of effect sizes in all of the analyses. Simply presenting results as significant or not is not as informative as presenting, for example, by what percent fecundity increased. From the figures, although fecundity did increase, it did so very so slightly that the biological significance of this temporary effect is not clear.

271: Why would this be a low leaf quality effect if leaves were being replaced?

286-288: Why is time now presented as continuous, when all of the other analyses use categories. Is the test presented on line 288 only for the comparison at month four? How can this be justified? That is cherry picking one time point to make the comparisons in lieu of an overall effect?

Lines 301: I would be more speculative here. Really, no mechanism can be established on the basis of the current study design. I would suggest instead stating this as a hypothesis that can be further investigated with additional data. It's intriguing for sure, but too speculative.

Review form: Reviewer 2

Recommendation

Major revision is needed (please make suggestions in comments)

Scientific importance: Is the manuscript an original and important contribution to its field?

Excellent

General interest: Is the paper of sufficient general interest?

Excellent

Quality of the paper: Is the overall quality of the paper suitable?

Marginal

Is the length of the paper justified?

No

Should the paper be seen by a specialist statistical reviewer?

No

Do you have any concerns about statistical analyses in this paper? If so, please specify them explicitly in your report.

No

It is a condition of publication that authors make their supporting data, code and materials available - either as supplementary material or hosted in an external repository. Please rate, if applicable, the supporting data on the following criteria.

Is it accessible?

Yes

Is it clear?

Yes

Is it adequate?

Yes

Do you have any ethical concerns with this paper?

No

Comments to the Author

The authors address important question regarding how homogeneity and heterogeneity of habitats influence adaptation. Specifically, they evaluate if heterogeneous environments with a benign and an unfavourable habitat together facilitate use of and adaptation to the unfavourable habitat. At the same time, they evaluate whether having just one species or interspecific competition alter these dynamics. It is a nice experiment with interesting results. There are some problems with the writing and the interpretation, however, one of which is linked to a flaw, as this reviewer sees it, in the experimental design, It's not a fatal flaw, but does limit inference.

The writing, as described below, is verbose at times, and not direct as it could be. I deeply sympathize with writing in a foreign language – such a challenge! But even so, the writing needs work.

There are two major parts of the interpretation that are troubling. The first is the references to stepping stone models and all that implies. The experiment was set up as experimental units of habitat that were totally unconnected. Some of those units were homogeneous, and some were heterogeneous. There was no migration whatsoever between them. This is simply not anywhere close to a stepping stone model. The units of replication had plants that were presumably close enough to each other that the authors did not bother to report a distance. Spider mites can move massive distances even without aerial dispersal and so can easily be making active choices of where to spend time, sampling the habitat as they go. It seems quite inappropriate to liken this set-up to a stepping stone model.

The second troubling area is the interpretation of competition, and this is linked to the absence of appropriate control or comparison treatments. The authors initiated populations of the focal species as two different sizes – 52 individuals (from some 13 isofemale lines) for single species units and 26 individuals (from those same 13 isofemale lines) for the 2 species (interspecific competition) units. They then attribute differences they observe to competition. To their credit, they mention drift due to smaller population size, but attribute that smaller population size to competition. That is a wild stretch. The smaller population size is, most parsimoniously, due to the set-up of the experiment itself. Without experimental controls of founding 26 individuals without heterospecifics, or founding with 52 individuals with heterospecifics, it is impossible to attribute the evident founder effect to the competitors, much less “ghost competition”.

In addition to these two more major items, line edits are suggested below, ranging from v. minor to more major.

14 – should read “The data WERE collected...” (not was)

In the abstract and elsewhere, “adaptation to” would be better than “adaptation towards”

The first sentence of the introduction doesn't quite make sense.

81 – I would think of this as evolutionary rescue, not genetic rescue.

104 “towards” should be replace with “to”

104 “the presence of” should be deleted

105 As noted above, here and elsewhere when referring to adaptation, “towards” should be replaced with “to”

134 “induced” does not work here. “Created” or “made” would be more appropriate.

134 “by different host plant treatments” implies that the plants are treated in different ways (for example, with or without fertilizers, or with or without induction of defenses by prior herbivory). I think what the authors mean is “by using different host plants”

134 “resource” should be plural. Host plants provide resources to the mites.

140 “to cross” should be replaced with “from crossing”

142 “to work from previous work” would read better as “to be effective”

146 “an equal number of 52 adult females” is quite confusing. I think the authors mean “52 adult females”.

156 this sentence needs work

169 In this paragraph and elsewhere, try to remove parenthetical statements. If the information is important, it should be incorporated into the text. If it is not, then it should be deleted. Only rarely does it make sense to provide information within parentheses.

177 This might read better as “We assessed evolutionary dynamics in performance by measuring fecundity.”

180 juvenile and maternal effects are always present and thus can’t be removed. A different description would work better. Perhaps “standardize”

212 build should be built, I think.

In the methods and results, population density is discussed before performance, and should be presented that way in the figures as well. Figure 1a should show population density, not performance.

Figure S1 is more informative than the current figure 1B and would prefer it in the manuscript. It is also not clear why there is no presentation of evolutionary responses to homogeneous and heterogeneous environments. The equivalent graph to S1 should be made for the performance data.

257 – control and experimental are not useful terms here – more descriptive would be to give the host plant names or to give a description like “benign versus suboptimal environment”

262 It would be nice if the writing could be a bit more active and descriptive. For example, “Populations from the suboptimal environment evolved higher fecundity on the cucumber host plant in four months.”

265 – “hostile” is too... hostile. Perhaps use “sink” or another descriptor, or alternatively provide evidence that the defenses in pepper have evolved overtime against spider mites in particular. If that is the case, then “hostile” might be an appropriate descriptor.

278 – as for the comment about 262, try to write more descriptively and actively. For example, “Fecundity on pepper decreased over time.” instead of “Fecundity tests performed on pepper showed an influence of time on the number of eggs laid by the mites.”

295 – similarly what is currently: “we found evidence for facilitation towards this challenging environment in a heterogeneous setting where more benign habitat (cucumber) was also present” could be stated more simply. For example, We found evidence that the presence of a benign habitat facilitated use of a challenging habitat.

This also brings up “environment” and “habitat” The authors should think about which they mean and stick to it.

320 – I continue to think this is more generally a potential example of evolutionary rescue rather than genetic rescue, if indeed it fits the “rescue” model at all.

341 – This idea is an interesting one, but since the competition treatment was started with half the

population size, it is impossible to say that competition leads to these potential drift effects. It is more likely that starting with a smaller population directly caused lower population fitness.

Figures 2 and 3 are difficult to interpret because the authors show box plots rather than model means and 95% confidence limits about those model means. It can be nice to have information about the raw data as well, but that could be in the form of a partially transparent spread of points in the background for each model mean and confidence limit.

Decision letter (RSPB-2019-0135.R0)

08-Feb-2019

Dear Ms Bisschop:

I am writing to inform you that your manuscript RSPB-2019-0135 entitled "Source-sink and transient local adaptation dynamics in an arthropod experimental metapopulation" has, in its current form, been rejected for publication in Proceedings B.

This action has been taken on the advice of referees, who have recommended that substantial revisions are necessary. With this in mind we would be happy to consider a resubmission, provided the comments of the referees are fully addressed. However please note that this is not a provisional acceptance.

Sincerely,

Proceedings B
mailto: proceedingsb@royalsociety.org

Associate Editor

Comments to Author:

Both reviewers were very positive about the aims and potential interest of this experimental study on adaptation in spider mites, and enjoyed reading the manuscript. However, both reviewers have also raised concerns about the experimental and statistical approach, and the interpretation and discussion of the results. Among their major comments, referee 1 makes recommendations about the analyses, including the use of effect sizes and presentation of full models to clearly show tests of the hypotheses. Referee 2 has important comments about the experimental set-up: both in terms of its description as a stepping stone model (given foraging or dispersal distances of the spider mites), and the need for abundance controls in the interpretation of the effects of interspecific competition. Both reviewers also recommend toning down some conclusions in the light of revised analyses, and provide a comprehensive set of very helpful constructive criticisms for the authors to address.

Reviewer(s)' Comments to Author:

Referee: 1

Comments to the Author(s)

In this ms, the authors use a lab-based experimental-evolution system to investigate how habitat heterogeneity (in quality and diversity) and competitive interactions can potentially interact to shape local adaptation in a meta-community context. They interpret their results to suggest that in heterogenous landscapes, adaptation towards stressful habitats may be critically shaped by both ecological dynamics via the contribution of benign habitats acting as source habitats amidst patches of sink habitats, as well as transient competitive and adaptive effects.

Overall comments:

The questions posed here are excellent and very much at the cutting edge of unresolved questions at the interface of ecology and evolutionary biology. Moreover, I really enjoyed reading this paper and it got me thinking about a number of issues in evolutionary ecology in a meta-population context.

However, I feel that the current version of the ms does not allow for a full evaluation of the results and that a number of additional issues may need to be considered further. Detailed comments are below. Overall, I felt that the presentation of the results needs to be re-worked and the decisions for the statistical analyses need to be further justified. At times, some of the results seem to be cherry-picked with the Discussion focusing on marginal effects that do not seem justified enough to be the focus. Additionally, the current presentation of the results relies too much on statistical significance rather than presentations of effect sizes. The observed effect sizes seem rather marginal and while statistically significant, do not appear biologically very meaningful – although to be fair I think the authors can address this given their knowledge of the study system and certainly even marginal increases in fitness are crucial for adaptive evolution. I do see merit in this study and found it fascinating, but I do feel that some work is needed to be able to better evaluate the work done here.

I hope that my comments are helpful to the authors in revising the presentation of their very interesting study.

Minor comments:

Line 61: Why resource partitioning? Seems that any environmental factor that could give a species a negative-frequency dependent advantage (e.g., resources, predators, mutualists) could be equally important.

Lines 64: Unclear what is meant by a sink becoming “preferred.” I can see how they could still be used, but how would a preference for them evolve through selection? Selection does not move populations down fitness surfaces.

81: I’d specify locally stabilizing mechanism, because regionally it could be stabilizing.

93: This is not universally true and would only occur if traits conferring competitive fitness advantages are aligned in the direction of adaptive evolution for traits driving local adaptation to some new habitats.

195: What are the bracketed values?

207 and 221: I do not understand the rationale for a model selection procedure here. The goal does not seem to be to predict performance per se or to determine what factors contribute most to performance. Rather, the goal is to test a hypothesis that a set of factors and their interactions can have an effect on performance. As such, I would prefer to see the full model for any comparisons. There seems to be no need for model selection procedures in this study. I might be misunderstanding this, but the full two (intra and interspecific competition) by four (control, pepper, cuk, pepp/cuk) factorial model design needs to be implemented in the statistical model and those results presented. Planned contrasts between the core comparisons could then be done.

211: I do not see how intraspecific competition per se was assessed here. To do so would require manipulating density and showing that as density increases fecundity declines. Perhaps this is known from previous experiments or other work in this system, but it is critical to demonstrate competition, rather than simply assuming it. Perhaps the authors simply mean effect of interspecific competition.

218: Unless there is a gross issue with non-linearity in plant deterioration rates, treating time as a category and the resulting many multiple comparisons among time categories, rather than a slope seems odd to me. It seems that knowledge of the study system should dictate this “possible” non-linear effect among the plant species.

241-: I realize that there was extinction on the pepper plant, but unless this occurred immediately, it would be useful to see the same figure as figure one, but for the homogeneous treatments as well; e.g., Figure 1B should include the homogenous treatments. This would allow for a nice visualization. Figure S1 is close and should be moved to the main text, but include the homo pepper as well. This would also provide an illustration for the points made on Lines 249-255.

In figure 1, why does time begin at 50 days? Why not at the start of the introductions (e.g., day 0)?

Note: For all figures – please avoid red and green for choice of colors. Quite a few folks are color blind and these colors cannot be distinguished.

Shouldn’t figure 2 y axes be labelled fecundity as well?

254: It's confusing to call one taxa an "interspecific competitor" because they are both interspecific competitors. Please state which went extinct.

258: Please give some measures of effect sizes in all of the analyses. Simply presenting results as significant or not is not as informative as presenting, for example, by what percent fecundity increased. From the figures, although fecundity did increase, it did so very slightly that the biological significance of this temporary effect is not clear.

271: Why would this be a low leaf quality effect if leaves were being replaced?

286-288: Why is time now presented as continuous, when all of the other analyses use categories. Is the test presented on line 288 only for the comparison at month four? How can this be justified? That is cherry picking one time point to make the comparisons in lieu of an overall effect?

Lines 301: I would be more speculative here. Really, no mechanism can be established on the basis of the current study design. I would suggest instead stating this as a hypothesis that can be further investigated with additional data. It's intriguing for sure, but too speculative.

Referee: 2

Comments to the Author(s)

The authors address important question regarding how homogeneity and heterogeneity of habitats influence adaptation. Specifically, they evaluate if heterogeneous environments with a benign and an unfavourable habitat together facilitate use of and adaptation to the unfavourable habitat. At the same time, they evaluate whether having just one species or interspecific competition alter these dynamics. It is a nice experiment with interesting results. There are some problems with the writing and the interpretation, however, one of which is linked to a flaw, as this reviewer sees it, in the experimental design, It's not a fatal flaw, but does limit inference.

The writing, as described below, is verbose at times, and not direct as it could be. I deeply sympathize with writing in a foreign language – such a challenge! But even so, the writing needs work.

There are two major parts of the interpretation that are troubling. The first is the references to stepping stone models and all that implies. The experiment was set up as experimental units of habitat that were totally unconnected. Some of those units were homogeneous, and some were heterogeneous. There was no migration whatsoever between them. This is simply not anywhere close to a stepping stone model. The units of replication had plants that were presumably close enough to each other that the authors did not bother to report a distance. Spider mites can move massive distances even without aerial dispersal and so can easily be making active choices of where to spend time, sampling the habitat as they go. It seems quite inappropriate to liken this set-up to a stepping stone model.

The second troubling area is the interpretation of competition, and this is linked to the absence of appropriate control or comparison treatments. The authors initiated populations of the focal species as two different sizes – 52 individuals (from some 13 isofemale lines) for single species units and 26 individuals (from those same 13 isofemale lines) for the 2 species (interspecific competition) units. They then attribute differences they observe to competition. To their credit, they mention drift due to smaller population size, but attribute that smaller population size to competition. That is a wild stretch. The smaller population size is, most parsimoniously, due to the set-up of the experiment itself. Without experimental controls of founding 26 individuals

without heterospecifics, or founding with 52 individuals with heterospecifics, it is impossible to attribute the evident founder effect to the competitors, much less “ghost competition”.

In addition to these two more major items, line edits are suggested below, ranging from v. minor to more major.

14 – should read “The data WERE collected...” (not was)

In the abstract and elsewhere, “adaptation to” would be better than “adaptation towards”

The first sentence of the introduction doesn’t quite make sense.

81 – I would think of this as evolutionary rescue, not genetic rescue.

104 “towards” should be replace with “to”

104 “the presence of” should be deleted

105 As noted above, here and elsewhere when referring to adaptation, “towards” should be replace with “to”

134 “induced” does not work here. “Created” or “made” would be more appropriate.

134 “by different host plant treatments” implies that the plants are treated in different ways (for example, with or without fertilizers, or with or without induction of defenses by prior herbivory).

I think what the authors mean is “by using different host plants”

134 “resource” should be plural. Host plants provide resources to the mites.

140 “to cross” should be replaced with “from crossing”

142 “to work from previous work” would read better as “to be effective”

146 “an equal number of 52 adult females” is quite confusing. I think the authors mean “52 adult females”.

156 this sentence needs work

169 In this paragraph and elsewhere, try to remove parenthetical statements. If the information is important, it should be incorporated into the text. If it is not, then it should be deleted. Only rarely does it make sense to provide information within parentheses.

177 This might read better as “We assessed evolutionary dynamics in performance by measuring fecundity.”

180 juvenile and maternal effects are always present and thus can’t be removed. A different description would work better. Perhaps “standardize”

212 build should be built, I think.

In the methods and results, population density is discussed before performance, and should be presented that way in the figures as well. Figure 1a should show population density, not performance.

Figure S1 is more informative than the current figure 1B and would prefer it in the manuscript. It is also not clear why there is no presentation of evolutionary responses to homogeneous and heterogenous environments. The equivalent graph to S1 should be made for the performance data.

257 – control and experimental are not useful terms here – more descriptive would be to give the host plant names or to give a description like “benign versus suboptimal environment”

262 It would be nice if the writing could be a bit more active and descriptive. For example, “Populations from the suboptimal environment evolved higher fecundity on the cucumber host plant in four months.”

265 – “hostile” is too... hostile. Perhaps use “sink” or another descriptor, or alternatively provide evidence that the defenses in pepper have evolved overtime against spider mites in particular. If that is the case, then “hostile” might be an appropriate descriptor.

278 – as for the comment about 262, try to write more descriptively and actively. For example, “Fecundity on pepper decreased over time.” instead of “Fecundity tests performed on pepper showed an influence of time on the number of eggs laid by the mites.”

295 – similarly what is currently: “we found evidence for facilitation towards this challenging environment in a heterogeneous setting where more benign habitat (cucumber) was also present” could be stated more simply. For example, We found evidence that the presence of a benign habitat facilitated use of a challenging habitat.

This also brings up “environment” and “habitat” The authors should think about which they mean and stick to it.

320 – I continue to think this is more generally a potential example of evolutionary rescue rather than genetic rescue, if indeed it fits the “rescue” model at all.

341 – This idea is an interesting one, but since the competition treatment was started with half the population size, it is impossible to say that competition leads to these potential drift effects. It is more likely that starting with a smaller population directly caused lower population fitness.

Figures 2 and 3 are difficult to interpret because the authors show box plots rather than model means and 95% confidence limits about those model means. It can be nice to have information about the raw data as well, but that could be in the form of a partially transparent spread of points in the background for each model mean and confidence limit.

Author's Response to Decision Letter for (RSPB-2019-0135.R0)

See Appendix A.

RSPB-2019-0738.R0

Review form: Reviewer 1

Recommendation

Accept with minor revision (please list in comments)

Scientific importance: Is the manuscript an original and important contribution to its field?

Excellent

General interest: Is the paper of sufficient general interest?

Good

Quality of the paper: Is the overall quality of the paper suitable?

Good

Is the length of the paper justified?

Yes

Should the paper be seen by a specialist statistical reviewer?

No

Do you have any concerns about statistical analyses in this paper? If so, please specify them explicitly in your report.

No

It is a condition of publication that authors make their supporting data, code and materials available - either as supplementary material or hosted in an external repository. Please rate, if applicable, the supporting data on the following criteria.

Is it accessible?

Yes

Is it clear?

Yes

Is it adequate?

Yes

Do you have any ethical concerns with this paper?

No

Comments to the Author

I was original referee 1. Overall, the authors have done a fine job addressing my original set of comments on their ms. I find the work interesting and although there are some limitations, all studies have them and the authors are largely clear on these limitations in inference.

It's quite interesting that the effects of interspecific competition did not depend on host. I would have imagined that being initially mal-adapted would have greatly accentuated the effects of interspecific competition. Perhaps this suggests that maladaptation buffers the effects of interspecific competition during colonization a bit. So, the opposite of expectations. Regardless, this is a very cool finding.

A few more minor points to consider:

Lines 49: The phrasing 'per definition' is a bit awkward, perhaps simply change 'per' to 'by'

Line 56 and elsewhere: change "which are usually having" to "which often have." This and the previous are but a few examples where a bit more attention to some of the writing would help to improve the readability. I won't provide such line-by line comments for the remainder of the ms, but I'd encourage the authors to go through the ms and look for similar places in the writing that could be improved.

Lines 72: if standing genetic variation is indefinitely maintained, then adaptation won't happen. Maybe the authors mean maintain genetic variation in a meta-population context?

Line 92: More of a comment; but note that lower population sizes when invading could be advantageous in reducing competition if the invaders are ecological distinct; e.g., negative frequency dependence is operating.

Line 138: It would also potentially act as an ecological source, correct?

Line 254: The sentence beginning "Evidently..." can be removed

Line 328: True, but is this not in the context of temporal environmental heterogeneity? That is, gradual adaptation (e.g., a non-rapid loss of genetic variance) would allow population to potentially adapt to fluctuating environmental conditions, or temporally changing environmental conditions?

Line 355: This seems like a bit of an overstatement, at least in the sense that the experiment was not particularly long-term. Moreover, adaptation to pepper never occurred, so while it may afford a potential opportunity for adaptation to extreme environments, this was still never achieved.

Line 365: While perhaps true, the results from this study also suggest that patch quality matters. If patches are just poor-quality habitat (pepper) then they have no real benefit and only costs. So, it remains important to consider both.

Figures:

I can appreciate the aesthetics, and a trend to make figures more artistic, but I would suggest that the inset figures of the plotted plants be removed. They add nothing to presenting the data and, if anything, make the figures more cluttered and difficult to interpret.

Similarly, presenting the cross bars and 95%CI superimposed above the violin plots is a bit much. I would remove the violin plots, or perhaps just have a point and line for the mean and CI, as the box is rather large.

Review form: Reviewer 2

Recommendation

Major revision is needed (please make suggestions in comments)

Scientific importance: Is the manuscript an original and important contribution to its field?

Excellent

General interest: Is the paper of sufficient general interest?

Excellent

Quality of the paper: Is the overall quality of the paper suitable?

Good

Is the length of the paper justified?

Yes

Should the paper be seen by a specialist statistical reviewer?

No

Do you have any concerns about statistical analyses in this paper? If so, please specify them explicitly in your report.

No

It is a condition of publication that authors make their supporting data, code and materials available - either as supplementary material or hosted in an external repository. Please rate, if applicable, the supporting data on the following criteria.

Is it accessible?

Yes

Is it clear?

Yes

Is it adequate?

Yes

Do you have any ethical concerns with this paper?

No

Comments to the Author

The authors did a nice job in their revision. On a second read, the mismatch of parts of the introduction to the paper as a whole stood out. The title says the paper is about metapopulations, but there aren't really metapopulations. In fact the term metapopulation isn't used in the entire introduction or conclusion, but there's a large focus on source-sink dynamics that seems a bit misplaced since movement was unlimited. There are good and poor host plants grown together with complete mixing between them. Can these truly be considered source-sink populations? Rather, the paper is really about how habitat heterogeneity vs homogeneity and competition influence adaptation. It is important that the introduction focus more explicitly on environmental heterogeneity and competition, and that the conclusion do the same (the discussion of the role of competition could be expanded a bit.). The research remains important and elegant – the writing in the introduction should more clearly reflect the work done, however.

A few minor suggestions follow.

Line 87 - could you provide a concrete example instead of just a citation?

Line 99 - "rarely" implies there are some studies. they should definitely be cited. What do those studies find? A brief overview of that pertinent literature should be provided

In the methods, an illustration of the experimental design would be really helpful for the reader. Magalhaes 2014 in JEB has a nice example.

Line 201 - Performance is just the number of eggs laid? That is fecundity. An environment can be more attractive for oviposition, but can be poor with respect to emergence or survival.

Line 311 - per the comment above, fecundity isn't necessarily the same as performance, thus I'm not fully convinced of adaptation. If you have data to link fecundity more closely to fitness, that would be more convincing.

Review form: Reviewer 3

Recommendation

Major revision is needed (please make suggestions in comments)

Scientific importance: Is the manuscript an original and important contribution to its field?

Excellent

General interest: Is the paper of sufficient general interest?

Excellent

Quality of the paper: Is the overall quality of the paper suitable?

Good

Is the length of the paper justified?

Yes

Should the paper be seen by a specialist statistical reviewer?

No

Do you have any concerns about statistical analyses in this paper? If so, please specify them explicitly in your report.

No

It is a condition of publication that authors make their supporting data, code and materials available - either as supplementary material or hosted in an external repository. Please rate, if applicable, the supporting data on the following criteria.

Is it accessible?

Yes

Is it clear?

Yes

Is it adequate?

Yes

Do you have any ethical concerns with this paper?

No

Comments to the Author

.

Decision letter (RSPB-2019-0738.R0)

29-Apr-2019

Dear Ms Bisschop:

Your manuscript has now been peer reviewed and the reviews have been assessed by an Associate Editor. The reviewers' comments (not including confidential comments to the Editor) and the comments from the Associate Editor are included at the end of this email for your reference. As you will see, the reviewers and the Editors have raised some concerns with your manuscript and we would like to invite you to revise your manuscript to address them.

Research ethics:

Use of animals and field studies:

Please submit a copy of your revised paper within three weeks. If we do not hear from you within this time your manuscript will be rejected. If you are unable to meet this deadline please let us know as soon as possible, as we may be able to grant a short extension.

Best wishes,

Proceedings B
mailto: proceedingsb@royalsociety.org

Associate Editor:

Comments to Author:

Both original referees were enthusiastic about the resubmission of this experimental study on the dynamics of spider mite adaptation to host plants. The main remaining recommendations of both referees refer to the text and presentation of the manuscript. Please see Referee 2's comments regarding the use of the term "metapopulation" in the title (but not elsewhere in the manuscript), and regarding the focus of the Introduction. Please also see Referee 1's comments, including those referring to the figures, and the clarity and polishing of language through the manuscript.

Reviewer(s)' Comments to Author:

Referee: 1

Comments to the Author(s).

I was original referee 1. Overall, the authors have done a fine job addressing my original set of comments on their ms. I find the work interesting and although there are some limitations, all studies have them and the authors are largely clear on these limitations in inference.

It's quite interesting that the effects of interspecific competition did not depend on host. I would

have imagined that being initially mal-adapted would have greatly accentuated the effects of interspecific competition. Perhaps this suggests that maladaptation buffers the effects of interspecific competition during colonization a bit. So, the opposite of expectations. Regardless, this is a very cool finding.

A few more minor points to consider:

Lines 49: The phrasing 'per definition' is a bit awkward, perhaps simply change 'per' to 'by'

Line 56 and elsewhere: change "which are usually having" to "which often have." This and the previous are but a few examples where a bit more attention to some of the writing would help to improve the readability. I won't provide such line-by-line comments for the remainder of the ms, but I'd encourage the authors to go through the ms and look for similar places in the writing that could be improved.

Lines 72: if standing genetic variation is indefinitely maintained, then adaptation won't happen. Maybe the authors mean maintain genetic variation in a meta-population context?

Line 92: More of a comment; but note that lower population sizes when invading could be advantageous in reducing competition if the invaders are ecological distinct; e.g., negative frequency dependence is operating.

Line 138: It would also potentially act as an ecological source, correct?

Line 254: The sentence beginning "Evidently..." can be removed

Line 328: True, but is this not in the context of temporal environmental heterogeneity? That is, gradual adaptation (e.g., a non-rapid loss of genetic variance) would allow population to potentially adapt to fluctuating environmental conditions, or temporally changing environmental conditions?

Line 355: This seems like a bit of an overstatement, at least in the sense that the experiment was not particularly long-term. Moreover, adaptation to pepper never occurred, so while it may afford a potential opportunity for adaptation to extreme environments, this was still never achieved.

Line 365: While perhaps true, the results from this study also suggest that patch quality matters. If patches are just poor-quality habitat (pepper) then they have no real benefit and only costs. So, it remains important to consider both.

Figures:

I can appreciate the aesthetics, and a trend to make figures more artistic, but I would suggest that the inset figures of the plotted plants be removed. They add nothing to presenting the data and, if anything, make the figures more cluttered and difficult to interpret.

Similarly, presenting the cross bars and 95%CI superimposed above the violin plots is a bit much. I would remove the violin plots, or perhaps just have a point and line for the mean and CI, as the box is rather large.

Referee: 2

Comments to the Author(s).

The authors did a nice job in their revision. On a second read, the mismatch of parts of the introduction to the paper as a whole stood out. The title says the paper is about metapopulations, but there aren't really metapopulations. In fact the term metapopulation isn't used in the entire introduction or conclusion, but there's a large focus on source-sink dynamics that seems a bit misplaced since movement was unlimited. There are good and poor host plants grown together with complete mixing between them. Can these truly be considered source-sink populations? Rather, the paper is really about how habitat heterogeneity vs homogeneity and competition influence adaptation. It is important that the introduction focus more explicitly on environmental heterogeneity and competition, and that the conclusion do the same (the discussion of the role of competition could be expanded a bit.). The research remains important and elegant – the writing in the introduction should more clearly reflect the work done, however.

A few minor suggestions follow.

Line 87 - could you provide a concrete example instead of just a citation?

Line 99 - "rarely" implies there are some studies. they should definitely be cited. What do those studies find? A brief overview of that pertinent literature should be provided

In the methods, an illustration of the experimental design would be really helpful for the reader. Magalhaes 2014 in JEB has a nice example.

Line 201 - Performance is just the number of eggs laid? That is fecundity. An environment can be more attractive for oviposition, but can be poor with respect to emergence or survival.

Line 311 - per the comment above, fecundity isn't necessarily the same as performance, thus I'm not fully convinced of adaptation. If you have data to link fecundity more closely to fitness, that would be more convincing.

Author's Response to Decision Letter for (RSPB-2019-0738.R0)

See Appendix B.

Decision letter (RSPB-2019-0738.R1)

03-Jun-2019

Dear Ms Bisschop

I am pleased to inform you that your manuscript RSPB-2019-0738.R1 entitled "Transient local adaptation and source-sink dynamics in experimental populations experiencing spatially heterogeneous environments." has been accepted for publication in Proceedings B.

The AE has recommended publication, but also suggests some minor revisions to your manuscript. Therefore, I invite you to respond to the comments and revise your manuscript. Because the schedule for publication is very tight, it is a condition of publication that you submit the revised version of your manuscript within 7 days. If you do not think you will be able to meet this date please let us know.

In order to ensure effective and robust dissemination and appropriate credit to authors the dataset(s) used should be fully cited. To ensure archived data are available to readers, authors should include a 'data accessibility' section immediately after the acknowledgements section.

This should list the database and accession number for all data from the article that has been made publicly available, for instance:

[http://datadryad.org/submit?journalID=RSPB&manu=\(Document not available\)](http://datadryad.org/submit?journalID=RSPB&manu=(Document%20not%20available)) which will take you to your unique entry in the Dryad repository. If you have already submitted your data to dryad you can make any necessary revisions to your dataset by following the above link. Please see <https://royalsociety.org/journals/ethics-policies/data-sharing-mining/> for more details.

Sincerely,

Dr Sasha Dall
 Editor, Proceedings B
<mailto:proceedingsb@royalsociety.org>

Associate Editor:

Board Member

Comments to Author:

The authors have carefully addressed and justified their responses to the previous round of revisions. There are just a few typos to correct:

Line 75 - please rephrase as "maintain the genetic variation needed to fuel"

Line 110 - please insert "mites" after "spider"

Line 129 - write "five-week-old" to correspond to the terminology used on the previous line

Line 297 - remove the comma after "island"

Line 368 - "interactions" in plural

Author's Response to Decision Letter for (RSPB-2019-0738.R1)

See Appendix C.

Decision letter (RSPB-2019-0738.R2)

03-Jun-2019

Dear Ms Bisschop

I am pleased to inform you that your manuscript entitled "Transient local adaptation and source-sink dynamics in experimental populations experiencing spatially heterogeneous environments." has been accepted for publication in Proceedings B.

Your article has been estimated as being 9 pages long. Our Production Office will be able to confirm the exact length at proof stage.

Open Access

Paper charges

Sincerely,

Appendix A

Dear Editor,

We thank you and the reviewers for the very useful and constructive suggestions in the review and addressed all of the reviewers' comments in the revised version. Below, the replies to the comments are added in blue. We feel that our changes have led to a significantly improved manuscript and hope that the paper is now suitable for publication in Proceedings of the Royal Society B.

Yours sincerely,

Karen Bisschop (on behalf of all authors).

Reviewer(s)' Comments to Author:

Referee: 1

Comments to the Author(s)

In this ms, the authors use a lab-based experimental-evolution system to investigate how habitat heterogeneity (in quality and diversity) and competitive interactions can potentially interact to shape local adaptation in a meta-community context. They interpret their results to suggest that in heterogeneous landscapes, adaptation towards stressful habitats may be critically shaped by both ecological dynamics via the contribution of benign habitats acting as source habitats amidst patches of sink habitats, as well as transient competitive and adaptive effects.

Overall comments:

The questions posed here are excellent and very much at the cutting edge of unresolved questions at the interface of ecology and evolutionary biology. Moreover, I really enjoyed reading this paper and it got me thinking about a number of issues in evolutionary ecology in a meta-population context.

We thank the reviewer for these very kind words and are glad he/she enjoyed reading the manuscript.

However, I feel that the current version of the ms does not allow for a full evaluation of the results and that a number of additional issues may need to be considered further. Detailed comments are below. Overall, I felt that the presentation of the results needs to be re-worked

and the decisions for the statistical analyses need to be further justified. At times, some of the results seem to be cherry-picked with the Discussion focusing on marginal effects that do not seem justified enough to be the focus. Additionally, the current presentation of the results relies too much on statistical significance rather than presentations of effect sizes. The observed effect sizes seem rather marginal and while statistically significant, do not appear biologically very meaningful – although to be fair I think the authors can address this given their knowledge of the study system and certainly even marginal increases in fitness are crucial for adaptive evolution. I do see merit in this study and found it fascinating, but I do feel that some work is needed to be able to better evaluate the work done here.

We thank the reviewer for the completeness of his review and addressed the points that were raised in the comments below.

I hope that my comments are helpful to the authors in revising the presentation of their very interesting study.

The comments have improved the manuscript a lot and we are very grateful for that.

Minor comments:

Line 61: Why resource partitioning? Seems that any environmental factor that could give a species a negative-frequency dependent advantage (e.g., resources, predators, mutualists) could be equally important.

We agree with the reviewer that resource partitioning is not the only way species could persist, we rephrased it as: “In novel heterogeneous environments, this persistence is maintained by mechanisms responding to negative frequency-dependency such as resource partitioning where...”.

Lines 64: Unclear what is meant by a sink becoming “preferred.” I can see how they could still be used, but how would a preference for them evolve through selection? Selection does not move populations down fitness surfaces.

We agree with the reviewer and changed “preferred” into “used”.

81: I’d specify locally stabilizing mechanism, because regionally it could be stabilizing.

We changed it to a local stable rescue mechanism.

93: This is not universally true and would only occur if traits conferring competitive fitness advantages are aligned in the direction of adaptive evolution for traits driving local adaptation to some new habitats.

We added this requirement to the statement in the manuscript: “and if traits for adaptation towards new habitats and for higher competitive fitness are aligned”.

195²: What are the bracketed values?

The values were indicating the 95% confidence interval and the outcome of the linear model to investigate potential differences in the percentage of zeros per month. We agree with the reviewer that this was not clear in the initial manuscript and added some more information.

207 and 221: I do not understand the rationale for a model selection procedure here. The goal does not seem to be to predict performance per se or to determine what factors contribute most to performance. Rather, the goal is to test a hypothesis that a set of factors and their interactions can have an effect on performance. As such, I would prefer to see the full model for any comparisons. There seems to be no need for model selection procedures in this study.

We thank the reviewer for his/her concerns. As this is an experiment, we indeed tested hypothesis on the determinants of mite performance in relation to population structure, host plant and time since the start of the experiment. As typical in ecological experiments, we came up with predictions regarding the main effects, but simultaneously needed to explore to which degree the main effects showed additive and interactive effects. Obviously, and even if we would have several clear predictions, we cannot use independent data and perform the multiple tests here. We therefore followed the (in ecology & evolution) accepted method of model inference. This can be done by backwards procedure from the full model, but then decisions need to be made on (i) the sequence of elimination of the explanatory variables and (ii) the cut-off level to decide a variable to be kept. We, however, followed the more elegant way of model selection (see Johnson & Omland 2004 TREE - but also many other discussions in the scientific literature), in which all competing models are compared and the best one(s) maintained for interpretation. This approach leads to a much more balanced and sound interpretation of the results and avoids a priori steering of the statistical results (the p-values) based on arbitrary criteria. We do provide the full model results in appendix as these can be seen as the reference for all models.

I might be misunderstanding this, but the full two (intra and interspecific competition) by four (control, pepper, cuk, pepp/cuk) factorial model design needs to be implemented in the statistical model and those results presented. Planned contrasts between the core comparisons could then be done.

We understand that this full factorial model would be interesting for core comparisons, but we did not implement a control for intra- and interspecific competition in our design. It is therefore impossible to implement this model in the manuscript.

211: I do not see how intraspecific competition per se was assessed here. To do so would require manipulating density and showing that as density increases fecundity declines. Perhaps this is known from previous experiments or other work in this system, but it is critical to demonstrate competition, rather than simply assuming it. Perhaps the authors simply mean effect of interspecific competition.

We thank the reviewer for pointing out this mistake in our manuscript and changed it to the effect of interspecific competition.

218: Unless there is a gross issue with non-linearity in plant deterioration rates, treating time as a category and the resulting many multiple comparisons among time categories, rather than a slope seems odd to me. It seems that knowledge of the study system should dictate this “possible” non-linear effect among the plant species.

We feel that time as a categorical effect would be more appropriate with our study system for several reasons. First of all, we cannot assume a linear response of fitness to time; adaptation may occur in jumps, especially when triggered by periodic immigration events. Secondly, we cannot assume that plant quality was constant or linearly changing with time. The results in the control population showed indeed a change in time. We added some more information to explain this better: “Time was treated as a categorical variable because we cannot assume a linear response of adaptation and leaf quality because leaf quality for the fecundity tests might change in time.”

241-: I realize that there was extinction on the pepper plant, but unless this occurred immediately, it would be useful to see the same figure as figure one, but for the homogeneous treatments as well; e.g., Figure 1B should include the homogenous treatments. This would allow for a nice visualization. Figure S1 is close and should be moved to the main text, but include the homo pepper as well. This would also provide an illustration for the points made on Lines 249-255.

We understand the comment of the reviewer, but cannot provide results of the populations on homogeneous pepper as we were not able to measure the fecundity. We explain this in the revision. We did include the fecundity data of the homogeneous cucumber islands in Fig. 1 and incorporated Fig. S1 into this figure.

In figure 1, why does time begin at 50 days? Why not at the start of the introductions (e.g., day 0)?

The first fecundity was measured after two months, therefore we are unable to start with fecundity at day 0. We noticed that not enough information was provided in the methods and

added “We sampled five *T. urticae* females from each plant species per island at 2, 4, 6, 8 and 10 months during the experiment.”

Note: For all figures – please avoid red and green for choice of colors. Quite a few folks are color blind and these colors cannot be distinguished.

We thank the reviewer for this suggestion and changed the colours into green and purple as this should be better for people with colour blindness.

Shouldn't figure 2 y axes be labelled fecundity as well?

We changed the labels on the y axes on Fig. 2 to “fecundity”.

254: It's confusing to call one taxa an “interspecific competitor” because they are both interspecific competitors. Please state which went extinct.

We added the species name to the interspecific competitor.

258: Please give some measures of effect sizes in all of the analyses. Simply presenting results as significant or not is not as informative as presenting, for example, by what percent fecundity increased. From the figures, although fecundity did increase, it did so very so slightly that the biological significance of this temporary effect is not clear.

We included effect sizes for the reported statistics in the Results as the percent fecundity increase.

271: Why would this be a low leaf quality effect if leaves were being replaced?

Although we tried to standardise as much as possible by buying the same variety of seeds, keeping a temperature between 25-30°C and a fixed light regime of 16:8 LD, we cannot guarantee the same initial leaf quality in a long-term experiment. Differences in potting soil or humidity can for instance cause differences. These differences did not cause effects between treatments in the experiment itself as plants that were grown together were divided amongst the islands, but because performance tests were done through time and we need plants of two, three or four weeks old (for bean, cucumber and pepper respectively), we could not provide plants from the same batch for each time point. To clarify this better, we rephrased it in the sentence: “All populations, including the control population, had a lower performance in the last month which seems to be due to differences in the leaf used per fecundity tests”.

286-288: Why is time now presented as continuous, when all of the other analyses use categories. Is the test presented on line 288 only for the comparison at month four? How can this be justified? That is cherry picking one time point to make the comparisons in lieu of an overall effect?

We thank the reviewer for pointing out that it was not clear. Time is also a categorical variable in the analyses for this paragraph, we therefore rephrased the sentence to: “But, we did detect a

transient signal of local adaptation. Mites sampled from pepper plants had a significantly higher fecundity on pepper than the mites sampled from cucumber from the same island at four months”. The test presented here is thus a comparison of the different treatments at a certain time point and not through time.

We detected a transient difference in Fig. 1 between mites from pepper and cucumber and wanted to statistically test this. As this was only temporary, we decided to look at the comparisons within time instead of for an overall effect over time.

Lines 301: I would be more speculative here. Really, no mechanism can be established on the basis of the current study design. I would suggest instead stating this as a hypothesis that can be further investigated with additional data. It’s intriguing for sure, but too speculative.

We agree that we must be clear that our explanation is speculative as it cannot be validated based on our results. We rephrased it: “It is likely that these source-sink dynamics initially created demographic rescue but also allowed for genetic adaptation. If so, this evolutionary rescue could have been only transient, because further expansion of source populations broke down adaptation through an influx of unadapted genes (i.e. genetic load). However, this hypothesis cannot be validated based on our research; further investigation with additional data is therefore necessary.”

Referee: 2

Comments to the Author(s)

The authors address important question regarding how homogeneity and heterogeneity of habitats influence adaptation. Specifically, they evaluate if heterogeneous environments with a benign and an unfavourable habitat together facilitate use of and adaptation to the unfavourable habitat. At the same time, they evaluate whether having just one species or interspecific competition alter these dynamics. It is a nice experiment with interesting results. There are some problems with the writing and the interpretation, however, one of which is linked to a flaw, as this reviewer sees it, in the experimental design, It’s not a fatal flaw, but does limit inference.

The writing, as described below, is verbose at times, and not direct as it could be. I deeply sympathize with writing in a foreign language – such a challenge! But even so, the writing needs work.

We thank the reviewer for the complement and for all the useful corrections in this revision, we did our best to improve the writing to make it more succinct.

There are two major parts of the interpretation that are troubling.

1. The first is the references to stepping stone models and all that implies. The experiment was set up as experimental units of habitat that were totally unconnected. Some of those units were homogeneous, and some were heterogeneous. There was no migration whatsoever between them. This is simply not anywhere close to a stepping stone model. The units of replication had plants that were presumably close enough to each other that the authors did not bother to report a distance. Spider mites can move massive distances even without aerial dispersal and so can easily be making active choices of where to spend time, sampling the habitat as they go. It seems quite inappropriate to liken this set-up to a stepping stone model.

We thank the reviewer for this insight and apologise for the wrong terminology in the initial manuscript. We agree that our experimental set-up did not meet the requirements for a spatial stepping stone model as the plants in the heterogeneous units were in direct contact with each other and none of the islands were connected. We added in the experimental set-up in the Methods part that plants within islands were in direct contact with each other to address the accurate comment of the reviewer. We also changed ‘stepping stone’ in the manuscript to ‘evolutionary stepping stone’ to distinct the term from spatial stepping stones and to make it clear that we interpret it as an intermediate step for adaptation. This terminology has for instance been used in Covert and other (2013) in PNAS.

2. The second troubling area is the interpretation of competition, and this is linked to the absence of appropriate control or comparison treatments. The authors initiated populations of the focal species as two different sizes – 52 individuals (from some 13 isofemale lines) for single species units and 26 individuals (from those same 13 isofemale lines) for the 2 species (interspecific competition) units. They then attribute differences they observe to competition. To their credit, they mention drift due to smaller population size, but attribute that smaller population size to competition. That is a wild stretch. The smaller population size is, most parsimoniously, due to the set-up of the experiment itself. Without experimental controls of founding 26 individuals without heterospecifics, or founding with 52 individuals with heterospecifics, it is impossible to attribute the evident founder effect to the competitors, much less “ghost competition”.

We agree with the reviewer that the interpretation of competition is too strong given that we are not providing an appropriate control in this manuscript, although we believe no such control exists, because we cannot keep population size and level of competition constant

at the same time. We therefore rephrased the paragraph in the discussion by adding that “the initial difference in the experimental set-up for the population sizes between both treatments, could have caused strong drift effects as well”.

In addition to these two more major items, line edits are suggested below, ranging from v. minor to more major.

3. 14 – should read “The data WERE collected...” (not was) In the abstract and elsewhere, “adaptation to” would be better than “adaptation towards”

We changed “adaptation towards” to “adaptation to” in the entire manuscript and also made data plural.

4. The first sentence of the introduction doesn’t quite make sense.

We rephrased the first sentence to “Local adaptation is an evolutionary solution allowing organisms to deal with changing environments, but it is subject to constraints imposed by the local population and the environment”.

5. 81 – I would think of this as evolutionary rescue, not genetic rescue.

We agree with the reviewer that an evolutionary rescue is more appropriate in this context and hence changed it in the manuscript.

6. 104 “towards” should be replace with “to”

As mentioned below the third comment, we replaced all the “adaptation towards” to “adaptation to” in the manuscript.

7. 104 “the presence of” should be deleted

We deleted those words.

8. 105 As noted above, here and elsewhere when referring to adaptation, “towards” should be replace with “to”

We adjusted this in the entire document.

9. 134 “induced” does not work here. “Created” or “made” would be more appropriate.

We changed “induced” to “created”.

10. 134 “by different host plant treatments” implies that the plants are treated in different ways (for example, with or without fertilizers, or with or without induction of defenses by prior herbivory). I think what the authors mean is “by using different host plants”

We thank the reviewer for this explanation and changed it in the sentence.

11. 134 “resource” should be plural. Host plants provide resources to the mites.

We changed it to plural as correctly noted by the reviewer.

12. 140 “to cross” should be replaced with “from crossing”

We changed “to cross” to “from crossing”.

13. 142 “to work from previous work” would read better as “to be effective”

We rephrased it as suggested by the reviewer.

14. 146 “an equal number of 52 adult females” is quite confusing. I think the authors mean “52 adult females”.

We changed our verbose way of writing to the less confusing suggestion of the reviewer.

15. 156 this sentence needs work

We agreed that the sentence was too long and complicated and changed it to: “Every week, we assessed the deterioration of the plants in the islands. When necessary, the two oldest plants were replaced by two new plants. This way of refreshment guaranteed enough time for the mites to move towards the fresh plants and also for a generation of mites to develop.”

16. 169 In this paragraph and elsewhere, try to remove parenthetical statements. If the information is important, it should be incorporated into the text. If it is not, then it should be deleted. Only rarely does it make sense to provide information within parentheses.

We thank the reviewer for this comment. We incorporated all parenthetical statements in the text or removed them when unnecessary.

17. 177 This might read better as “We assessed evolutionary dynamics in performance by measuring fecundity.”

We rephrased this sentence.

18. 180 juvenile and maternal effects are always present and thus can't be removed. A different description would work better. Perhaps “standardize”

We changed “remove” by “standardise”.

19. 212 build should be built, I think.

We changed the present form to the past form to be congruent with the text.

20. In the methods and results, population density is discussed before performance, and should be presented that way in the figures as well. Figure 1a should show population density, not performance.

Figure S1 is more informative than the current figure 1B and would prefer it in the manuscript. It is also not clear why there is no presentation of evolutionary responses to homogeneous and heterogeneous environments. The equivalent graph to S1 should be made for the performance data.

We agree with the reviewer that the visualisation would be improved by changing the figures. We therefore replaced Fig. 1 in the manuscript with Fig. S1 and added the performance data as well. In this way, the homogeneous cucumber data and the

heterogeneous data are presented together, both for the ecological and the evolutionary dynamics. We also changed the order of the panel because the ecological data are first discussed as correctly noted by the reviewer.

21. 257 – control and experimental are not useful terms here – more descriptive would be to give the host plant names or to give a description like “benign versus suboptimal environment”

We changed the title in the results as suggested by the reviewer.

22. 262 It would be nice if the writing could be a bit more active and descriptive. For example, “Populations from the suboptimal environment evolved higher fecundity on the cucumber host plant in four months.”

We adjusted the writing style in the Results to a more active and descriptive style.

23. 265 – “hostile” is too... hostile. Perhaps use “sink” or another descriptor, or alternatively provide evidence that the defenses in pepper have evolved overtime against spider mites in particular. If that is the case, then “hostile” might be an appropriate descriptor.

As no specific defences against spider mites have evolved, we agree with the reviewer and used “sink” instead.

24. 278 – as for the comment about 262, try to write more descriptively and actively. For example, “Fecundity on pepper decreased over time.” instead of “Fecundity tests performed on pepper showed an influence of time on the number of eggs laid by the mites.”

We rephrased the sentence as suggested by the reviewer.

25. 295 – similarly what is currently: “we found evidence for facilitation towards this challenging environment in a heterogeneous setting where more benign habitat (cucumber) was also present” could be stated more simply. For example, we found evidence that the presence of a benign habitat facilitated use of a challenging habitat.

We changed the sentence to make it less complicated. The new sentence is the same as proposed by the reviewer, but we changed “benign” to “more benign” to make a difference with the really benign bean host plant.

26. This also brings up “environment” and “habitat”. The authors should think about which they mean and stick to it.

We agree with the reviewer that this may cause confusion and changed ‘environment’ to ‘habitat’ in the entire manuscript. The only times we did not rephrase is was when it was used in a specific term, e.g. isolation by environment and environmental filter.

27. 320 – I continue to think this is more generally a potential example of evolutionary rescue rather than genetic rescue, if indeed it fits the “rescue” model at all.

We changed this to evolutionary rescue as well.

28. 341 – This idea is an interesting one, but since the competition treatment was started with half the population size, it is impossible to say that competition leads to these potential drift effects. It is more likely that starting with a smaller population directly caused lower population fitness.

As mentioned above in the major comments, we rephrased the interpretation of competition to: “Although ghost interactions are known to affect performance by a reduction in effective population size, and hence an increase in drift effects, the initial difference in the experimental set-up for the population sizes between both treatments, could have caused strong drift effects as well”.

29. Figures 2 and 3 are difficult to interpret because the authors show box plots rather than model means and 95% confidence limits about those model means. It can be nice to have information about the raw data as well, but that could be in the form of a partially transparent spread of points in the background for each model mean and confidence limit.

We agree with the reviewer that the figures were difficult to interpret. Novel figures are made with the violin plots representing the observed data and the crossbars showing the model means and their 95% confidence intervals.

Appendix B

Dear Editor,

We thank you and the reviewers for the constructive evaluation of our submitted manuscript. We are happy to hear that the referees were enthusiastic about the previous resubmission. We addressed all of the reviewers' comments in the revised version. Below, the replies to the comments are added in blue. We feel that our changes have led to a significantly improved manuscript and hope that the paper is now suitable for publication in Proceedings of the Royal Society B.

Yours sincerely,

Karen Bisschop (on behalf of all authors).

Associate Editor:

Both original referees were enthusiastic about the resubmission of this experimental study on the dynamics of spider mite adaptation to host plants. The main remaining recommendations of both referees refer to the text and presentation of the manuscript. Please see Referee 2's comments regarding the use of the term "metapopulation" in the title (but not elsewhere in the manuscript), and regarding the focus of the Introduction. Please also see Referee 1's comments, including those referring to the figures, and the clarity and polishing of language through the manuscript.

We are happy for this chance to revise the manuscript and agree with the main concerns of the editor and reviewers. We avoided the use of the term 'metapopulation' as it was indeed not applicable in our study as correctly noticed by the reviewer. We adjusted the figures as suggested by the reviewer and agree that this improves the clarity of the manuscript. We furthermore substantially revised the language in order to improve both the clarity and narrative of the manuscript and thank the reviewers for their suggestions.

Reviewer(s)' Comments to Author:

Referee: 1

Comments to the Author(s).

I was original referee 1. Overall, the authors have done a fine job addressing my original set of comments on their ms. I find the work interesting and although there are some limitations, all studies have them and the authors are largely clear on these limitations in inference.

We thank the reviewer for these kind words and are happy that we addressed the previous comments satisfactorily. Furthermore, we really appreciate the recognition of the values of our work and its limitations.

It's quite interesting that the effects of interspecific competition did not depend on host. I would have imagined that being initially mal-adapted would have greatly accentuated the effects of interspecific competition. Perhaps this suggests that maladaptation buffers the effects of interspecific competition during colonization a bit. So, the opposite of expectations. Regardless, this is a very cool finding.

We are very glad that the reviewer pointed out the interesting finding related to interspecific competition. We decided to elaborate more on the results of competition in the discussion and added: "It is surprising that we found no significant differences in the effects of competition on fecundity assessed on cucumber and pepper plants, as we expected maladaptation to new hosts to increase the effects of interspecific competition."

A few more minor points to consider:

Lines 49: The phrasing 'per definition' is a bit awkward, perhaps simply change 'per' to 'by'

We agree with the reviewer and rephrased the sentence.

Line 56 and elsewhere: change "which are usually having" to "which often have." This and the previous are but a few examples where a bit more attention to some of the writing would help to improve the readability. I won't provide such line-by-line comments for the remainder of the ms, but I'd encourage the authors to go through the ms and look for similar places in the writing that could be improved.

We apologise for the unnecessarily complicated writing and went through the manuscript to improve the readability. We thank the reviewer for pointing this out.

Lines 72: if standing genetic variation is indefinitely maintained, then adaptation won't happen. Maybe the authors mean maintain genetic variation in a meta-population context?

We agree with the reviewer that this sounds counterintuitive and rephrased it as suggested: "Connectedness may facilitate this local adaptation if standing genetic variation is maintained in connected but less challenging habitats,..."

Line 92: More of a comment; but note that lower population sizes when invading could be advantageous in reducing competition if the invaders are ecological distinct; e.g., negative frequency dependence is operating.

The fact that lower population sizes could be advantageous as well under for instance negative frequency dependence is indeed true and we are happy that the reviewer is pointing this out. We added a sentence in the discussion "Also, when negative frequency dependence is operating, invading under lower population sizes could be beneficial if invaders are ecologically distinct."

Line 138: It would also potentially act as an ecological source, correct?

We agree that the cucumber plant can also present an ecological source and that it should have been mentioned here, so we rephrased it as: "Cucumber is the more palatable of the two and hence

hypothesised to present an ecological source and a possible evolutionary stepping-stone to adaptation to pepper.”

Line 254: The sentence beginning “Evidently...” can be removed

We agree that the sentence is unnecessary and removed it from the manuscript.

Line 328: True, but is this not in the context of temporal environmental heterogeneity? That is, gradual adaptation (e.g., a non-rapid loss of genetic variance) would allow population to potentially adapt to fluctuating environmental conditions, or temporally changing environmental conditions?

We agree with the reviewer that the temporal change in habitat in our study is not completely parallel with the temporal environmental in the study we are referring to. We do think however that the comparison we made with our study is valid. We found that an abrupt change in habitat from the ancestral population on bean to pepper plants was too strong and none of the populations was able to survive. In combination with cucumber plants, a more gradual shift was made. We deem this gradual adaptation to be important to allow the establishment of the population and potential adaptation.

Line 355: This seems like a bit of an overstatement, at least in the sense that the experiment was not particularly long-term. Moreover, adaptation to pepper never occurred, so while it may afford a potential opportunity for adaptation to extreme environments, this was still never achieved.

We thank the reviewer to point this out to us. We rephrased the first sentence in the conclusion as followed: “Our study demonstrates transient adaptation in colonisation processes with episodes of higher potential for adaptation to extreme environments via a combination of ecological and evolutionary rescue. Persistent adaptation was however never achieved, presumably because it was eventually completely overruled by spill-over.”

Line 365: While perhaps true, the results from this study also suggest that patch quality matters. If patches are just poor-quality habitat (pepper) then they have no real benefit and only costs. So, it remains important to consider both.

We are glad that the reviewer highlights this consideration. Therefore we rephrased our final conclusion to: “As populations are not able to establish in homogenous landscapes consisting of marginal habitat, persistence can be ensured in intermediately connected landscapes where some patches of good quality may serve as enhancers of local adaptation to more marginal habitat.”

Figures:

I can appreciate the aesthetics, and a trend to make figures more artistic, but I would suggest that the inset figures of the plotted plants be removed. They add nothing to presenting the data and, if anything, make the figures more cluttered and difficult to interpret.

We thank the reviewer for the concerns and removed the plotted plants inside the figures. We do think that the small plant might help to understand the figure and we, therefore, placed it on top of the plot where it is not overlapping with the data.

Similarly, presenting the cross bars and 95%CI superimposed above the violin plots is a bit much. I would remove the violin plots, or perhaps just have a point and line for the mean and CI, as the box is rather large.

We agree with the reviewer that the figures need to be clear and not too dense. Therefore we followed the advice to change the crossbars to a point and line for the mean and CI. We did not remove the raw data in the figure as we deem it important to report this. Over the recent years, several groups have advocated the presentation of the actual data in graphs instead of data summaries (Drummond and Vowler 2011, Weissgerber et al. 2015, Rousselet et al. 2017, Postma and Goedhart 2019).

Referee: 2

Comments to the Author(s).

The authors did a nice job in their revision. On a second read, the mismatch of parts of the introduction to the paper as a whole stood out. The title says the paper is about metapopulations, but there aren't really metapopulations. In fact the term metapopulation isn't used in the entire introduction or conclusion, but there's a large focus on source-sink dynamics that seems a bit misplaced since movement was unlimited. There are good and poor host plants grown together with complete mixing between them. Can these truly be considered source-sink populations? Rather, the paper is really about how habitat heterogeneity vs homogeneity and competition influence adaptation. It is important that the introduction focus more explicitly on environmental heterogeneity and competition, and that the conclusion do the same (the discussion of the role of competition could be expanded a bit.). The research remains important and elegant – the writing in the introduction should more clearly reflect the work done, however.

We are glad that the reviewer approved our previous changes. We furthermore thank the reviewer for indicating the problem with the term “metapopulation” and changed the title to “Transient local adaptation and source-sink dynamics in experimental populations experiencing spatially heterogeneous environments.”

We feel that source-sink theories are applicable to our research and provided some more information: “The distribution of species is not restricted to areas where expected fitness is positive. While dispersing, species also come into contact with marginal and even completely unsuitable habitat but may reach substantial local population sizes there via spill-over effects. Species may therefore be ecologically rescued in marginal habitat by source-sink dynamics.” We think that considering the heterogeneous islands in our study as source-sink populations is appropriate, given that the pepper plants in our research are unsuitable as seen in the homogeneous islands consisting of this plant species. The mites found on this unsuitable habitat are spilling-over from the preferred cucumber plants.

A few minor suggestions follow.

Line 87 - could you provide a concrete example instead of just a citation?

We added a sentence to provide a concrete example: “An example of a general stress response is fluctuating temperatures leading to thermal generalism.”

Line 99 - "rarely" implies there are some studies. They should definitely be cited. What do those studies find? A brief overview of that pertinent literature should be provided

We thank the reviewer for noticing this, but think that so far no other study investigated the interplay between environmental heterogeneity and competition empirically, we rephrased it as: “Environmental heterogeneity and competition are thus anticipated to affect the process of local adaptation to novel habitats, but to our knowledge, they have not been simultaneously studied experimentally.”

In the methods, an illustration of the experimental design would be really helpful for the reader. Magalhaes 2014 in JEB has a nice example.

We agree that visualising the experimental design would be helpful for the reader and created an extra figure S1. We decided to place it in the supplementary information given the length of our manuscript, but we can add it in the main text if preferred by the editor.

Line 201 - Performance is just the number of eggs laid? That is fecundity. An environment can be more attractive for oviposition, but can be poor with respect to emergence or survival.

We changed “performance” in the sentence where the comment is referring to in “fecundity”, but feel comfortable to say that fecundity might be used as a proxy for adaptation in *Tetranychus urticae*. Please also see the reply on the comment below.

Line 311 - per the comment above, fecundity isn't necessarily the same as performance, thus I'm not fully convinced of adaptation. If you have data to link fecundity more closely to fitness, that would be more convincing.

We thank the reviewer for this concern and gave some more information about fecundity as a proxy for adaptation in the manuscript, “We chose fecundity as a proxy of adaptation because previous research demonstrated it to be the best predictor of adaptation compared to survival or development (Magalhães et al., 2007; Alzate et al., 2017, 2018).”

Appendix C

Dear Editor,

We thank you for the constructive evaluation of our submitted manuscript and are very happy to hear that our manuscript has been accepted for publication. We addressed your final comments in the revised version. Below, you can find our reply and the manuscript with tracked changes.

Yours sincerely,

Karen Bisschop (on behalf of all authors).

Comments to Author:

The authors have carefully addressed and justified their responses to the previous round of revisions. There are just a few typos to correct:

Line 75 - please rephrase as "maintain the genetic variation needed to fuel"

Line 110 - please insert "mites" after "spider"

Line 129 - write "five-week-old" to correspond to the terminology used on the previous line

Line 297 - remove the comma after "island"

Line 368 - "interactions" in plural

We thank the associate editor for noticing the writing errors and corrected these in the revised manuscript.